# Perceived Barriers and Facilitators to Breaking Up Sitting Time among Desk-Based Office Workers: A Qualitative Investigation Using the TDF and COM-B

**DOI:** 10.3390/ijerph16162903

**Published:** 2019-08-14

**Authors:** Samson O. Ojo, Daniel P. Bailey, David J. Hewson, Angel M. Chater

**Affiliations:** 1Institute for Health Research, University of Bedfordshire, Luton LU1 3JU, Bedfordshire, UK; 2Institute for Sport and Physical Activity Research, School of Sport Science and Physical Activity, University of Bedfordshire, Polhill Avenue, Bedford MK41 9EA, Bedfordshire, UK

**Keywords:** sedentary behaviour, barriers, facilitators, desk-based employees, office workers, COM-B, TDF, Behaviour Change Wheel

## Abstract

High amounts of sedentary behaviour, such as sitting, can lead to adverse health consequences. Interventions to break up prolonged sitting in the workplace have used active workstations, although few studies have used behaviour change theory. This study aimed to combine the Theoretical Domains Framework (TDF) and the Capability, Opportunity, and Motivation to Behaviour system (COM-B) to investigate perceived barriers and facilitators to breaking up sitting in desk-based office workers. Semi-structured interviews with 25 desk-based employees investigated barriers and facilitators to breaking up sitting in the workplace. Seven core inductive themes were identified: *‘Knowledge-deficit sitting behaviour’, ‘Willingness to change’, ‘Tied to the desk’, ‘Organisational support and interpersonal influences’, ‘Competing motivations’, ‘Emotional influences’,* and *‘Inadequate cognitive resources for action’.* These themes were then deductively mapped to 11 of the 14 TDF domains and five of the six COM-B constructs. Participants believed that high amounts of sitting had adverse consequences but lacked knowledge regarding recommendations and were at times unmotivated to change. Physical and social opportunities were identified as key influences, including organisational support and height-adjustable desks. Future research should identify intervention functions, policy categories and behaviour change techniques to inform tailored interventions to change sitting behaviour of office workers.

## 1. Introduction

Sedentary behaviour is defined as a waking energy expenditure ≤1.5 metabolic equivalents (METs) while sitting, reclining or lying down [1]. High levels of sedentary behaviour has been identified as a risk factor for a number of cardiometabolic health issues, including cardiovascular disease, type 2 diabetes, and obesity [2,3,4]. The workplace has been identified as a high-risk site for excessive sitting time in many western countries [5,6], with evidence that office workers spend the majority of their working day sitting [7]. Desk-based workers may sit for up to 82% of their working hours [5]. 

Studies have shown that breaking up long periods of sitting with short bouts of standing, light and moderate-intensity activity can reduce cardiometabolic risk markers including postprandial glucose, insulin, triglycerides and blood pressure [8,9,10,11,12]. Numerous interventions have been designed to break up sitting, including environmental changes to the workplace, such as provision of height-adjustable desks to switch between siting and standing, treadmill desks, and cycling desks [13,14,15], computer prompts [16,17,18,19], policy changes, counselling and provision of information [20]. However, while some of these interventions appear to be effective in reducing workplace sitting time [17,21,22], results from studies have been inconsistent. In addition, those using active workstations provide low quality evidence and there is a dearth of literature on long-term use according to a Cochrane review [23]. For instance, a study investigating the effectiveness of standing ‘hot desks’ on sedentary work time has reported no change in overall sedentary behaviour [24]. Similarly, a Swedish cross-sectional observational study reported that height-adjustable desks only resulted in a 5% decrease in call-centre workers’ total sitting time [15]. Moreover, a study carried out in four different companies reported that 60% of male and female employees who recently received height-adjustable desks stated using the desks only once a month or less [25]. This suggests that access to a height-adjustable desk does not necessarily translate to using it to alternate between sitting and standing [26]. Similarly, the effectiveness of using treadmill desks to increase light physical activity among desk-based office workers was hampered by poor adherence due to drop-out, injury, work conflict and being out of office [27].

A possible reason for the lack of adherence to interventions could be that they were not tailored to meet the needs of the participants. Few studies have evaluated the perception of employees to reducing workplace sitting before developing interventions [28,29]. Cole et al. [30] explored the perceptions of professionals with respect to factors that might hinder or promote sedentary behaviour at work. This was necessary as it allowed the participants to be part of the processes involved in making decisions towards changing behaviour [31]. Future research should use appropriate behaviour change theory to develop tailored interventions that consider specific preferences, work practices and settings [30].

The use of behaviour-change theory and models have been recommended for many contexts including addiction, prevention of disease, and professional practice [32,33]. It is helpful to assess barriers to change prior to the tailoring of interventions [34]. The Behaviour Change Wheel (BCW) is a layered framework designed to enable developers of interventions to move from the analysis of a behavioural problem to intervention design informed by evidence [35]. This framework provides a systematic and transparent way of identifying intervention functions and policy categories that provoke change [35]. The Medical Research Council guidance [36] for developing complex interventions outlines that researchers need to have a theoretical understanding of probable process of change. This can be achieved by drawing on existing evidence and theory, and if necessary, supplemented with qualitative interviews with stakeholders and/or the target population [36]. The COM-B (Capability, Opportunity, Motivation—Behaviour) model is at the hub of the BCW and is used to determine what needs to change. The COM-B model establishes that human behaviour is a result of the interaction that exists between Capability, Opportunity and Motivation [35]. Capability refers to individual’s physical and psychological ability to enact the target behaviour, which usually involves having the required cognitive processing ability, such as attention, memory, knowledge, and skills. Motivation on the other hand refers to the belief systems that strengthen and guide behaviour such as beliefs about capability, intention, and outcome expectancies. It further encompasses habitual processes, and emotional responses. Opportunity points to other factors beyond the individual, both social and physical, which can influence the enablement of the behaviour [35]. Each of the three components can be further subdivided to capture important distinctions within the constructs. For instance, (1) ‘psychological capability’ has to do with the capacity to engage in essential thought processes, while (2) ‘physical capability’ is concerned with skills to execute the behaviour. In terms of opportunity, (3) ‘physical opportunity’ is what is afforded by the environment to enable the behaviour, while (4) ‘social opportunity’ centres around cultural and social influences. With respect to motivation, (5) ‘reflective motivation’ involves thinking with the head, which includes evaluations and intentions while (6) ‘automatic motivation’ involves emotions and habits [35]. This system does not place priority on any of the individual components, however, it provides a way of identifying to what extent changing one or more components could lead to the desired change in behaviour [35]. COM-B has been found to be an effective model in explaining physical activity behaviours [37], however, less is known of its ability to predict sedentary behaviour.

The Theoretical Domains Framework (TDF) [38] was designed to help with the understanding of the concept of behaviour theoretically to effectively target processes for change [32]. The TDF has been used as the basis for identifying factors that predict guideline adherence; for structuring an interview schedule to capture drivers of behaviour using framework analysis [39,40] and for structuring the presentation of data analysis [41,42]. However, the best approach to identifying the most appropriate domains to be targeted for intervention is still unclear [43]. The TDF has 14 domains which are; ‘Knowledge’, ‘Skills’, ‘Social/Professional Role and Identity’, ‘Beliefs about Capabilities’, ‘Optimism’, ‘Beliefs about Consequences’, ‘Reinforcement’, ‘Intentions’, ‘Goals’, ‘Memory, Attention and Decision Processes’, ‘Environmental Context and Resources’, ‘Social Influences’, ‘Emotions’, and ‘Behavioural Regulation’ [38].

More recently, research has begun to use the TDF alongside COM-B. Qualitative studies investigating smoking cessation, stroke rehabilitation, diabetes prevention and antibiotic prescribing behaviour [44,45,46,47,48] have adopted this approach. This enables the expansion of each of the components of COM-B, allowing for more detail by using the TDF alongside [44,45,46,47,48] creating a parsimonious arrangement of probable influencing determinants of behaviour [49].

There is only one known study relating to sedentary behaviour, that has used the combination of the TDF and COM-B to develop an intervention, entitled Stand More AT Work (SMArT). This intervention aimed to reduce workplace sitting time in desk-based National Health Service (NHS) hospital staff [49]. Within three months of intervention, the study found that workplace sitting time had reduced by 50.6 min and by 64.4 min at six months in addition to sustainable long-term improved job performance [50]. With sedentary behaviour research applying both the COM-B model and TDF still at infancy stage, it is difficult to ascertain the generalisability of the current evidence. Therefore, further studies that target employees in different settings with purely administrative roles and high levels of sitting time are warranted [51]. This current study will draw on both the COM-B model and TDF combined to investigate the perceived barriers and facilitators to breaking up and reducing workplace sitting time among desk-based office workers to inform the development of future interventions.

## 2. Materials and Methods

### 2.1. Design

This study consisted of qualitative semi-structured interviews with desk-based employees between April and July 2017. The interview approach was chosen rather than focus group discussion to ensure participants could express themselves freely without any undue influence from others [52].

### 2.2. Participant Recruitment

A purposive sampling technique was used to identify desk-based employees working for a local authority and university located in the East of England. Participants were identified based on a prior quantitative study that assessed the levels of sitting time of desk-based employees in these two organisations. Participants aged 18–65 years with no mental or musculoskeletal impairment that may inhibit compliance with a future intervention and with a self-reported minimum of 5½ hours of sitting per work day, which equates to a threshold of 75% of a working day, were eligible to participate. The study was advertised to staff using an email invite sent out to those who met the inclusion criteria. Thirty-two interviews were initially scheduled to take place across the two worksites, however, saturation was reached after 25 interviews, with no new information or themes observed in the data [53]. No incentive was offered for participation.

### 2.3. Materials

A semi-structured interview schedule was used to identify barriers and facilitators to the uptake of an intervention aimed at reducing workplace sitting in the target population. The interview schedule was developed by SO and AC, informed by the Theoretical Domains Framework (TDF) [38], and then mapped onto each of the components of the COM-B model, as illustrated in Table 1. The COM-B model is based on the notion that behaviour will only occur if an individual possesses both the physical and psychological Capability to enact the behaviour, they possess both the social and physical Opportunity to enable the behaviour and have the reflective Motivation to perform the behaviour while overcoming the automatic Motivation to not engage or to engage with another competing behaviour [35].

### 2.4. Procedure

The study was approved by the Institute for Health Research Ethics Committee at the University of Bedfordshire on the 4th April 2016 (IHREC610). The semi-structured face-face interviews were conducted by SO and began with a general introduction and explanation of what the interview entailed. Prior to questioning, participants were given a participant information sheet and a consent form to sign and were asked if they were happy for the interview to be audio-recorded. The duration of each interview was between 30 to 60 min, which is in line with that deemed to be appropriate for this type of research [54].

### 2.5. Data Analysis 

Demographic data collected during the interviews were summarised through the use of descriptive statistics. The recorded semi-structured interviews were anonymously transcribed verbatim then analysed using NVivo qualitative data analysis software (Version 10, QSR International, Melbourne, Australia). All transcription was performed by SO. Using a combination of thematic analysis [55] and framework Analysis [56], and following an iterative process, SO reviewed the transcripts through the stages of familiarization, developing a thematic framework, indexing, charting, mapping and interpretation. Phrases and sentences that were salient and referred to barriers and facilitators to breaking up sitting time in the workplace were inductively identified and assigned codes. Recurrent themes were presented and discussed with AC and the wider team, and final themes were identified. Inter-rater reliability of the final thematic analysis was assessed using 10% of the transcripts, which were independently coded by AC, with comparisons made to resolve discrepancies [57]. Each theme was then deductively mapped to the TDF domains and COM-B. Deductive mapping of themes was independently carried out by both SO and AC, giving consideration to the definitions of each of the components of COM-B [35] and TDF domains [38]. Any disagreement over domains was resolved through discussion. Using both thematic analysis and framework analysis allowed naturally identified themes to be determined and then allocated to pre-selected theoretically-driven domains to assist in answering the research question [58].

## 3. Results

### 3.1. Characteristics of Participants

Twenty-five desk-based employees aged 26–59 years (40.9 ± 10.8 years) were interviewed (17 females, 8 males). Eight of the 25 participants were from the council offices, while the remaining 17 were university employees. All participants are presented using pseudonyms [59]. The mean body mass index (BMI) of participants was 25.7 ± 3.5 kg/m^2^. Four participants were classified as obese (BMI ≥ 30), with 10 participants overweight (BMI ≥ 25 to 30), and 11 participants normal weight (BMI ≥ 18.5 to < 25). Ethnicity was mixed with White British/Other as dominant (*n* = 15), Black Caribbean/African/British (*n* = 3), Pakistani (*n* = 3), Asian (*n* =1), Greek (*n* = 1), Cypriot (*n* = 1) and Russian (*n* = 1).

### 3.2. Core Themes—Determinants of Breaking Up and Reducing Sitting Time in the Workplace (COM-B Behavioural Diagnosis)

Inductive thematic analysis led to the identification of seven core themes, which were able to explain desk-based employees’ sedentary behaviour in the workplace. These themes were (1) *‘Knowledge-deficit sitting behaviour’,* (2) *‘Willingness to change’,* (3) *‘Tied to the desk’,* (4) *‘Organisational support and interpersonal influences’,* (5) *‘Competing motivations’,* (6) *‘Emotional influences’, and* (7) *‘Inadequate cognitive resources for action’.* These themes were then deductively mapped to five of the six COM-B constructs (*Psychological Capability, Reflective Motivation, Automatic Motivation, Social Opportunity, Physical Opportunity [omitting Physical Capability]*) and 11 of the TDF domains (*Knowledge, Beliefs about capabilities, Intentions, Beliefs about consequences, Environmental context and resources, Social influences, Reinforcement, Social/Professional role and identity, Emotion, Memory, attention and decision processes, Behavioural regulation [omitting Skills, Optimism and Goals]).* They are presented with COM-B constructs and TDF domains *shown in brackets*, (*COM-B; [TDF])* and within the thematic map in Figure 1, with an overview in Table 2.

(1) ‘Knowledge-deficit sitting’ (Psychological Capability; [Knowledge])

Lack of knowledge regarding sitting guidelines was unanimously cited as a reason for prolonged sitting in the workplace.

*“Ahem, I really don’t know what the advice is about total sitting time, the only advice I’ve seen is getting up regularly, but that’s really more about screen time, time away from your computer screen. Not really aware of the general advice to the general population about sitting. Just that, it’s not good to sit for too long”*.(Eunice, 47)

*“I actually have no idea what the recommended sitting time would be for a workplace environment or what the expert advice is, the right of length of time”*.(Jake, 37)

Lack of knowledge of the possible consequences of prolonged sitting was also highlighted.

*“I don’t really know anything about the other half and the implications of sitting down all day.”*.(Eunice, 47)

However, a high number of participants knew their sitting at work was excessive, yet were also aware that knowledge does not always lead to action.

*“There’s a complete disconnect between what I do and what I know is good practice. Like I said, it’s not done deliberately even though I know, you know it’s good practice to get up and walk around”*.(David, 32)

*“Many times, I’ve had to stand up but wait for two seconds for my legs to wake up long enough before I walk along to the photocopier. My sitting time is horrendous. Knowing it and doing it are two different things”*.(Taher, 44)

(2) ‘Willingness to Change’ (Reflective Motivation; [Beliefs about capabilities, Intentions, Beliefs about consequences])

Participants expressed mixed feelings towards their intentions to changing their sedentary behaviour in the workplace. While many participants were quite confident about breaking up and reducing their sitting time, there were some who were not so sure due to situations beyond their control.

*“I am quite confident. I don’t think anyone will have a problem with that, like I said, when I have a chance, I do like to go for a little walk”*.(Marcos, 31)

*“I feel confident if I am able to control my work environment, but if I am in a meeting or in a presentation then I wouldn’t be able to”*.(Eunice, 47)

However, there was high variability with respect to beliefs about the health consequences of prolonged sitting, including weight gain, heart problems, lethargy, backaches, numbness, bad posture, tiredness, and mental drainage.

*“Muscle wastage, exhaustion, not being alert, painful both mentally and physically, painful in the sense that you are not able to regulate your body temperature”*.(Farouq, 30)

*“For me, it’s putting on weight and not just exercising enough so I can honestly say to you that if I came in 8am in the morning, sometimes I don’t leave the office; I just walk down the corridor to make a cup of coffee until I leave by 4pm - that has got to be bad!”*.(Vickie, 59)

Apart from a perceived negative impact on health, it was also reported that prolonged sitting might result in a loss of productivity and concentration.

*“I think that you become less productive because you are just sitting in the same spot so everything else is not firing quite the same way and eye strain, in particular with using the computer, and just not getting your heart rate going as much I suppose is another thing”*.(Julie, 37)

When considering how to address issues of knowledge, motivation and perceived consequences in a future intervention, participants felt that education would be the best place to start.

*“Umm, I think if there’s a better awareness both with staff and management, umm, and like I said earlier, I don’t know what the guidelines are, so I don’t even know whether I am sitting too much.... And I imagine managers would be the same, umm, I think it starts with education and if managers are aware they will encourage you maybe to take breaks to do something”*.(Tiana, 47)

(3) ‘Tied to the desk’ (Physical Opportunity; [Environmental context and resources])

Most participants stated that the nature of their job impels them to work sitting down, without any structured breaks as it requires working on a computer, typing, responding to queries via emails, and making and receiving telephone calls.

*“I think the general nature of my work mitigates against me being active at work because it’s essentially sitting at a computer or sitting interviewing students or sitting in meetings.”*.(Becky, 58)

Moreover, irrespective of mindset, restriction due to heavy workload was seen as a hindrance to breaking up sitting time at regular intervals.

*“Workload, theoretically I should be moving around different areas, looking at people, I have staff in different environments, I should be visiting them... People don’t choose to want to sit all day, I think it’s the work level that makes you sit all day at the desk. For crying out loud with my workload perhaps if I go for a coffee break I could probably do it on my laptop while I have my coffee.”*.(Jude, 57)

*“If they’re customer facing and if they’re really busy, they don’t always get a chance to actually go and leave the desk to take a break and walk around and get some light exercise”*.(Sophia, 26)

However, many believed they would be able to break up and reduce their sitting time by alternating between sitting and standing whilst carrying on working if they had access to height-adjustable desks.

*“If you have those, you know, adjustable desks, absolutely that would make a huge difference for sure rather than moving around… If I had a desk, like one of those, like I said we’ve got a couple in the office that you can raise or lower, you will be more inclined to be standing for most of the day than sitting for sure”*.(Abi, 40)

However, concerns were raised by some of the participants about the cost implication of a height-adjustable desk, and whether organisations would be willing to invest in these.

*“I’ve seen these desks where you can stand up and work; that will be fantastic! Knowing about the cost implication in it, financial instability, may be too strong a word, of the organisation. Ahem, I don’t think any money will be put into that kind of motivation, but that will be fantastic”*.(Eunice, 47)

Furthermore, some participants considered the use of height-adjustable desks in an open plan office to be potentially disruptive and awkward, and suggested having a separate room.

*“Yeah, I mean that would be quite good, you know maybe that, maybe have a separate room for it… I do understand that we are in a multi-disability environment… that could be awkward because if my desk could go up and down and this other person can’t because she is sat down, that could be you. The next desk to me is probably here, so my desk going up and down is gonna probably really piss her off”*.(Taher 44)

(4) ‘Organisational support and interpersonal influences’ (Social Opportunity; [Social influences])

Several employees highlighted fear of being judged and condemned by their fellow colleagues, and of being penalised by their manager as a reason for not breaking up their sitting behaviour.

*“The main inhibitor is if people think they will be penalised because they think they’ve not been productive. I think the motivation will just come from knowing that they are supported by the manager”*.(Eunice, 47)

*“It’s probably a matter of attitude because if you’re just walking around, people think you’re, you’re wasting time. What are you doing? You know, chatting to people when you should be sitting at your desk working? Erm, and I think that’s perhaps the biggest thing, is attitude towards you getting up and walking around”*.(Tina, 58)

However, it was revealed that the onus is not on individuals alone, but also on organisations. Participants were keen to break up their sitting if there was a policy or written statement that clearly shows that their organisations are supportive of the culture of micro-breaks at work and that there would not be any form of condemnation or judgment.

*“Culture needs to change, and I think that’s where a flexible working time frame of the work day would allow it more. When it is fixed, you have to leave at a certain time. It is not allowing the flexibility to then get up and have a break from work and come back, it might mean you have to work later to get your hours during the day”*.(Cheryl, 31)

*“The only way really is becoming some sort of policy. Health and safety policy where it’s implemented like a body in every office that reminds people”*.(Jude, 57)

(5) ‘Competing motivations’ (Automatic Motivation; [Reinforcement, Social/Professional role and identity])

Many participants identified existing habits and reinforced daily routines that inhibit them from breaking up their sitting time, some of which include eating at the desk and surfing the internet whilst sitting at the desk during lunch breaks.

*“I actually work through my lunch; that is not good but I do work through my lunch time…… but you might be just surfing the net to look at what’s on YouTube, such as the latest diet or, you know, whatever interests that you might see… checking social media and stuff like that and catching up with whatever BBC news stories might be”*.(Lilian, 55)

Several employees also reported laziness and lack of drive for taking regular breaks. Some of them said they delay getting up until they have so many things to do:

*“Erm... I don’t want to use this word but I’m going to say laziness. Do you know what I mean? There’s sometimes I think, “Ahh, I could, I could go downstairs” but I’ll wait until I’ve got more things I need to do downstairs and I’ll go just once. So, I think laziness is, is a large bit of it”*.(Precious, 50)

*“We are in an open office, so you don’t have to necessarily get up and talk to somebody; you can just holler, it’s laziness so yeah work environment we are in, is pretty lazy. It impacts quite a lot”*.(Louisa, 41)

On the other hand, around half the participants admitted that they can engage in less sedentary time if they have a strong mindset and are mentally disciplined to break the habit in relation to their sitting time.

*“I think it is more of a discipline thing actually. Right, you do need to get up, you need to stretch, you need to do something rather than just sitting here”*.(John, 27)

*“Yeah, I need my mindset to change basically, I need to have, I need to get myself motivated to do something even if it’s just making a point of going out every lunch time and just walking round the block”*.(Vickie, 59)

(6) ‘Emotional influences’ (Automatic Motivation; [Emotion, Reinforcement])

The feeling of not wanting to move was often linked to times when low in mood, while participants reported moving more often when they felt happy.

*“When I’m more cheerful, happier more relaxed then I’ll probably get up more because I will go and talk to somebody or spend a bit more time you know, not just sitting focusing on what I’m doing”*.(Julie, 37)

*“Yeah, erm... I guess if I was in a low mood I would be more inclined to sit still and not get up because I don’t want, I don’t perhaps necessarily want to find somebody, no-not find somebody, I wouldn’t want to bump into somebody who then is going to give me something I don’t want, in terms of work or a problem or something, so I would probably more likely if I was in a low mood to kind of hide up here and just, just kind of keep my head down and keep it out the way”*.(Precious, 50)

In contrast, some participants believed that their job, rather than their mood, determines their sitting behaviour:

*“No, not for me, my job determines my sitting behaviour and what I do determines my sitting behaviour but my mood doesn’t, no!”*.(Abi, 40)

*“Regardless of if am upset or am happy it doesn’t really affect how much I sit, if I have a requirement I will go. If I am allowed to move freely I would go, it gives me better mind”*.(Farouq, 30)

Some participants believed introducing some form of reward system (Reinforcement) for sitting less would have an influence on their mood, which in turn, could encourage them to take more breaks from sitting in the workplace.

*“I don’t know, I mean a reward system always seems to help with children; what would you do with an adult I don’t know.... yeah maybe incentives but I don’t know what that incentive would be… whether if you do this and you get a bag of apples at the end of the month; do you know what I mean”*.(Abi, 40)

*“It wouldn’t necessarily have to be money, it could be as I say, a kind of build credits for some sort of treat or, I don’t know, half an hour of you know, leave early one day or, you know, have some, some leave in lieu or, or something like that… Who wouldn’t be happy?”*.(Precious, 50)

(7) ‘Inadequate cognitive resources for action’ (Psychological Capability; [Memory, attention and decision processes, Behavioural regulation])

Participants also stated having inadequate cognitive resources that support breaking up sitting time. For instance, they blamed their sitting behaviour on being engrossed in their daily tasks, and as a result, they forget to take micro-breaks. Being immersed in work also led to a neglect of physiological or psychological triggers to break up sitting time in order to meet a deadline.

*“I get focused on work, erm, I don’t notice the time so if I am writing a research paper or if I am writing a key document I simply don’t notice the time.”*.(David, 32)

*“I think because I’m concentrating either writing about something or maybe doing something technical on the website, rather than break the concentration even when tired, I often stick at it till I get the job done. And sometimes that can take longer than you would think”*.(Eunice, 47)

However, most of the participants believed that installing an application that flashes on their computer, known as computer on-screen prompts, would serve as a reminder for them to get up and take a short break from their work.

*“Also doing some sort of algorithms for your computer, so that when your computer realises that you’ve been sat down it flashes up or something, a buzzer or something that tells me when I’ve sat still for X amount of time”*.(David, 32)

*“I can’t remember, but basically something that locks your screen and tells you, you have to move forward or something like that and then come back or my Fitbit, just charging it now… it’s up to the individual, because am pretty sure about 90% of my office would like to have that app”*.(Sophia, 26)

Despite the popularity, some participants believed computer on-screen prompts can easily be ignored or go unnoticed, especially when the users are in a meeting or working away from their desks and computers.

*“Hmm, it is about how you sit if you will see the prompt. Sometimes you can be in a meeting or board meeting, so you might not be aware of the prompt as you are not sitting close to your computer at that time, so it depends on the nature of the office and how often people sit with their PC”*.(Cheryl, 31)

It was also perceived that older employees might be irritated seeing their screens locked automatically without warning.

*“Might be a good one, but it might be annoying for other people. Like we obviously have old members in the team, so I think they might not appreciate locking their screens every half an hour”*.(Sophia, 26)

## 4. Discussion

Barriers to breaking up and reducing sitting time at work were attributed to the core themes of: ‘Knowledge-deficit sitting behaviour’, ‘Willingness to change’, ‘Tied to the desk’, ‘Organisational support and interpersonal influences’, ‘Competing motivations’, ‘Emotional influences’, and ‘Inadequate cognitive resources for action’. These themes were linked to five of the six COM-B elements, without strong evidence for Physical Capability barriers. This is likely due to the exclusion of people with musculoskeletal problems in the study design.

The use of the COM-B model [35] was instrumental in identifying an overview of factors contributing to prolonged sitting time in the workplace. Mapping the data further to the TDF [38] ensured an exhaustive behavioural diagnosis of the determinants of the behaviour. Eleven of the TDF domains (*Knowledge, Beliefs about capabilities, Intentions, Beliefs about consequences, Environmental context and resources, Social influences, Reinforcement, Social/Professional role and identity, Emotion, Memory, attention and decision processes, Behavioural regulation)* were highlighted deductively following the initial inductive thematic analysis, without strong evidence for *Skills, Optimism and Goals.* This may again be linked to the inclusion criteria (free from musculoskeletal ill health) and the nature of the interviews investigating barriers as opposed to issues that may highlight optimistic views or goal setting.

Participants often did not engage in breaking up sitting behaviour as they lacked the Psychological Capability in the form of knowledge or guidance for sitting at work and stated that it would be important for them to understand why it would be necessary. However, despite not knowing recommendations for sitting time, the participants still believed their sitting pattern was bad and showed positive intentions towards participating in a behaviour change intervention to break up sitting in the workplace. This is consistent with findings from the SMArT study that used focus groups to explore what needed to change to reduce NHS office-based employees’ sitting time based on questions developed from the COM-B model and TDF [49]. In this present study, participants’ knowledge was found to be an important TDF domain.

Both Physical and Social Opportunities were found to be crucial to breaking up and reducing sitting behaviour in the present study. This is consistent with work by Mackenzie et al. [60], in which most of the participants blamed the sedentary nature of their job, tight deadlines, peers’ influence, an unsupportive organisational culture, fear of being judged due to the open plan structure of their office that does not allow for any privacy, and an excessive workload, for their inability to break up sitting time. The participants in the current study felt it would be helpful to be provided with opportunity, which would require their work environment to be restructured, including the use of a height-adjustable desk, on-screen computer prompts, scheduled breaks, and adequate support from the organisational management. These findings are consistent with those identified in focus group discussions among employees and executives to identify strategies that could be used to influence workplace sedentary time in a study by De Cocker and colleagues [28]. However, concerns were raised about the cost implication of a height-adjustable desk and whether organisations would be willing to invest in such a scheme to enhance the Physical Opportunity to sit less. There is evidence that suggests the long-term indirect costs of employees’ presenteeism and absenteeism stemming from sedentary behaviour-associated illness might be higher than the cost of height-adjustable desks. For instance, height-adjustable desks cost ≥ £279 (US $375) per unit for a single display [61] and ≥ £305 (US $400) per unit for a dual display [62], which compares favourably with the indirect cost of presenteeism for businesses (which is far more than the cost of absenteeism) that was approximately £194 (US $255) per employee per annum [63]. Previous work has shown that organisational managers and employers are perceived as gatekeepers that determine the implementation of the recommendations to break up prolonged sitting in the workplace [64], which was also identified in the current study. Similarly, the social opportunity provided by being within a team or culture in which everybody (employees and managers) implemented the plan, was desirable, highlighting the importance of Social Opportunity to facilitate breaking up sitting behaviour.

Both Automatic and Reflective Motivation were also found to be key determinants of participants’ sitting behaviour and their intention to be involved in an intervention that aimed to break up sitting time. In terms of Automatic Motivation, regardless of opportunity, laziness, lack of drive and habitual behaviour such as eating at the desk could hinder employees from breaking up their sitting time. However, this could be mediated by their Reflective Motivation regarding beliefs about the consequences of breaking up sitting time, such as benefits to health. Automatic Motivation had a varied effect; mood, for instance, was found to either increase or decrease participants’ sitting behaviour, and sometimes, appeared to have no effect. However, motivation to increase engagement with breaking up sitting could arise from the introduction of a reward system. This suggests that reinforcement could be an important TDF domain to influence emotions and therefore should be taken into consideration during intervention development.

The findings from the analysis of the interviews in this study suggest that the drivers of behaviour could be either intrinsic or extrinsic. Intrinsic drivers are those that are personal to employees, while extrinsic drivers are determined by the environment and employers. However, this implies that breaking up sitting time will require a more complex intervention using concerted behavioural planning that involves both office workers and their employers, as found in a previous study [49]. It was evident that there is currently no such collaborative scheme, where employees are knowledgeable, ready, willing and able, and feel supported to break up their sitting in the workplace. To trigger a change, there is an urgent need for intervention developers to focus on educating office workers about risks attached to prolonged sitting and what is currently considered as acceptable sitting time durations during a working day. To ensure such a change occurs would require a culture of collaborative behavioural planning and the development of a complex intervention that would need to take Psychological Capability, Social and Physical Opportunity, and Reflective and Automatic Motivation into consideration. However, going by the ‘less is more’ principle of intervention design, only a fraction of target drivers of behaviour should be selected for rigorous intervention testing, instead of targeting several with no focus [65]. Subsequent studies on intervention development involving identification of intervention functions and active ingredients should be mindful of these considerations.

One of the limitations of qualitative studies is concern over generalisability due to a small sample size in comparison with quantitative studies and their subjectivity with regards to researchers’ interpretation. However, this study has provided an in-depth analysis of the barriers to breaking up sitting time at work and potential facilitators that can be targeted in intervention design. A further limitation is that physical activity of the participant was not considered during recruitment. However, as the focus of this study was on workplace sitting time, the time spent sitting during working hours was one of the criteria for inclusion rather than level of physical activity, which can be mutually exclusive. Future consideration should be given to occupational health surveillance and the identification of those at risk of excess sitting.

## 5. Conclusions

The present study took a novel approach adopting both the COM-B model and TDF to identify factors that could influence desk-based employee sitting behaviour, which ensured that the study was theoretically-underpinned and office worker-led, rather than researcher-led [66,67]. It also ensured that any future implementation would be person-centred [68]. Future research should take this behavioural diagnosis work and consider following the systematic approach provided by the developers of the COM-B model and Behaviour Change Wheel [35] to further identify intervention functions, policy categories and behaviour change techniques to support the development and examination of an intervention to change sitting behaviour of office workers.

## Figures and Tables

**Figure 1 ijerph-16-02903-f001:**
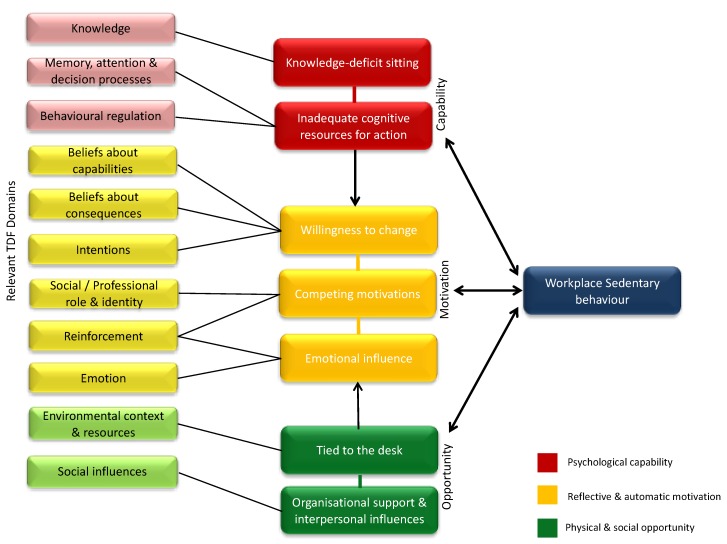
Qualitative themes linked to the Theoretical Domains Framework (TDF) and Capability, Opportunity and Motivation—Behaviour model (COM-B) as determinants of workplace sedentary behaviour.

**Table 1 ijerph-16-02903-t001:** Capability, Opportunity, and Motivation to Behaviour system (COM-B)/Theoretical Domains Framework (TDF)-informed interview schedule.

COM-B Construct	COM-B Micro-Construct	TDF Domain	Eliciting Questions
CAPABILITY	Psychological	Knowledge	*Can you start by telling me your understanding of current advice by experts about how much sitting time is okay?*
			*Prompt—At what point do you think sitting becomes too much?*
			*Prompt—What do you think are the consequences of sitting for long periods?*
			*How do you think your sitting time compares with this advice?*
		Memory, Attention and Decision Processes	*What do you think could help you overcome barriers that might prevent you from breaking up and reducing your sitting at work?*
		Behavioural Regulation	*What would need to change to help you break up long periods of sitting at work?*
	Physical	Skills	*What things might prevent you from breaking up and reducing your sitting time at work?*
OPPORTUNITY	Social	Social influences	*How do the people you work with influence your sitting time?*
			*How could your work colleagues and employer help you to break up and reduce your sitting time?*
	Physical	Environmental context and Resources	*How does your environment at work influence your sitting behaviour?*
			*How would your environment need to change to make breaking up and reducing sitting easier for you?*
MOTIVATION	Reflective	Beliefs about Capabilities Social/Professional Role & Identity	*How confident are you that you could break up your sitting time?*
		Beliefs about Consequences Optimism	*How much benefit do you feel breaking up your sitting time would give you?*
		Intentions Goals	*What would need to change to help you break up long periods of sitting at work?*
	Automatic	Emotion	*How do you think your mood during the day would influence your sitting patterns?*
			*What about your habits?*
		Reinforcement	*How could you overcome these to break up and reduce your sitting time at work?*

**Table 2 ijerph-16-02903-t002:** Combined COM-B and TDF analysis of the determinants of breaking up sitting.

COM-B Construct	COM-B Micro-Construct	TDF Domain	Perceived Barriers and Facilitators to Breaking Up Sitting Time
CAPABILITY	Psychological	Knowledge	**Barrier:** Unsure of the sitting guidelines recommended by experts
			**Facilitator:** Information on how often to break up sitting time
			**Facilitator:** Creating awareness of both the benefits of breaking up sitting and consequences of prolonged sitting
			**Facilitator:** Knowing alternative means to break up sitting (e.g., going to speak with colleagues instead of using intercom)
		Memory, Attention and Decision Processes	**Barrier:** Forget to take breaks from sitting because of being focused on work-task
			**Facilitator:** Computer on-screen prompts/cues to serve as a reminder to break up sitting at work
		Behavioural Regulation	**Barrier:** Lack of strategies to break up sitting time
			**Facilitator:** Self-devised strategies to break existing habits (for instance, stop eating at the desk, stand up and drink, regular comfort breaks, etc.)
	Physical	* Skills	**Barrier:** Physical health, i.e., ability to stand
OPPORTUNITY	Social	Social influences	**Barrier:** Conforming to colleagues’ sitting patterns
			**Barrier:** Organisational culture and climate do not support taking breaks (e.g., feeling of being watched and/or judged)
			**Facilitator:** Team support or buddy system
			**Facilitator:** Workplace ‘Get Active Officer’ demonstrating the behaviour
			**Facilitator:** Introduce walking or standing meetings
			**Facilitator:** Supportive organisational culture that encourages taking breaks and active work, to include senior management support
			**Facilitator:** Discourage tea making by work colleagues, encourage making it themselves
	Physical	Environmental context and Resources	**Barrier:** Heavy workload prevents taking breaks
			**Barrier:** Nature of the job and type of tasks impels sitting at work
			**Barrier:** Mixed perception of prompt: could be ignored
			**Barrier:** Open plan offices
			**Barrier:** Concerns for cost burden of height-adjustable desk
			**Facilitator:** Need a height-adjustable desk that allows working while standing
			**Facilitator:** Moving printers, water dispensers and toilet away from close proximity
			**Facilitator:** Use of a treadmill/stand up chairs or buzzing chairs
			**Facilitator:** Use of a hot standing desk
MOTIVATION	Reflective	Beliefs about Capabilities	**Barrier:** Confidence in ability to break up sitting time
		Beliefs about Consequences	**Barrier:** Lack of belief that taking a break will lead to positive consequences
		Intention	**Barrier:** State of contemplation or lack of determination to break up sitting time
		* Goal	**Facilitator:** Developing intentions, goals and strategies
		* Optimism	**Facilitator:** Engender an optimistic view of reducing sitting behaviour and outcome
	Automatic	Emotion	**Barrier:** Mood negatively influencing sitting time
			**Facilitator:** Find strategies to enhance emotional influence
		Reinforcement	**Barrier:** Bad habits inhibiting breaks (for instance, having lunch at the desk, surfing the internet)
			**Facilitator:** Promoting breaks through reward system

* Not highlighted as a core theme in this research.

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
