# Peer review of "Perceived Barriers and Facilitators to Breaking Up Sitting Time among Desk-Based Office Workers: A Qualitative Investigation Using the TDF and COM-B"

_ijerph, 2019, doi:10.3390/ijerph16162903_

Round 1

Reviewer 1 Report

This is a well thought-out and well written qualitative study, exploring barriers and facilitators to breaking up sitting time in office workers using the TDF and COM-B. I have a few suggestions that would strengthen the paper:

1)      The introduction requires more rationale on what behaviour change approaches such as TDF/COM-B are beneficial for public health interventions e.g evidence they increase effectiveness in PA/SB trials

2)      More detail on COM-B would be beneficial in the introduction – spend the time to break down the 6 components into slightly more detail and how they relate to behaviour.

3)      Methods: more detail is needed on how you mapped findings to COM-B and TDF, what resources did you use to guide this? The sentence in Results ‘led to the natural development of seven core themes’ is very vague, especially as insufficient detail on the process is given in the Methods.

4)      COM-B description in Materials section sounds like 5-components rather than 6-components – suggest you report this a bit more clearly

5)      Sometimes ‘breaking up sitting time’ and ‘active workstations’ are used interchangeably. It would be good to be clearer how these two terms are related

6)      Table 2 seems a bit excessive and not needed. If you added some more detail summarizing the population in the ‘characteristics of participants’ section I feel this would be sufficient e.g Ethnicity. Were participants PA levels assessed in any way?

7)      You reference that this qualitative work is part of a behaviour change intervention. It would be good to refer more explicitly to this intervention in the paper, especially in the next steps section of the Discussion.

8)      It could be clearer that personas were pseudonyms

Author Response

Dear Reviewer 1,

We would like to take this opportunity to thank you for your comments regarding our paper and time that you have taken to review our work.  Please find detailed replies to your comments below.

Best wishes

Dr Angel Chater on behalf of all co-authors

1)       The introduction requires more rationale on what behaviour change approaches such as TDF/COM-B are beneficial for public health interventions e.g. evidence they increase effectiveness in PA/SB trials

Response: Research using TDF/COM-B in the area of sedentary behaviour is limited.  However, Howlett et al (2017) has shown that the models can significantly predict physical activity behaviour. Moreover, qualitative studies investigating smoking cessation, stroke rehabilitation, diabetes prevention and antibiotic prescribing behaviour have adopted both TDF and COM-B in developing interventions that target these public health issues.  This has been made more visible in the introduction section Lines 93-95 and 106-108. It should be noted that COM-B and TDF represent layers of the Behaviour Change Wheel. The outcome of the only study (SMArt) that has tested the efficacy of the BCW for sedentary behaviour has been reported (see Line 111 – 115).

2)      More detail on COM-B would be beneficial in the introduction – spend the time to break down the 6 components into slightly more detail and how they relate to behaviour.

Response: Thank you. Details of how the six components of COM-B relate to behaviour have been provided in the introduction as seen in Line 76-95.

3)       Methods: more detail is needed on how you mapped findings to COM-B and TDF, what resources did you use to guide this? The sentence in Results ‘led to the natural development of seven core themes’ is very vague, especially as insufficient detail on the process is given in the Methods.

Response: Following inductive analysis for the identified themes within the interviews, deductive analysis then mapped these themes to the COM-B and TDF.  This mapping was individually carried out by both SO and AC by giving consideration to the definitions of each of the components of COM-B (Michie et al., 2011) and TDF domains (Cane et al., 2012). Disagreement over domains was resolved through discussion.  This has been made clearer in the text, see Line 160 – 170 in the method section.

4)       COM-B description in Materials section sounds like 5-components rather than 6-components – suggest you report this a bit more clearly.

Response: Thank you. This has been revised with added text to include the two elements of motivation. Please see Lines 145-146.

5)       Sometimes ‘breaking up sitting time’ and ‘active workstations’ are used interchangeably. It would be good to be clearer how these two terms are related.

Response: The use of active workstations is one of the many strategies that has been used to break up sitting time in the workplace. When active workstations have been mentioned, this is a specific technique used to break up sitting time. We have made this clearer.

6)       Table 2 seems a bit excessive and not needed. If you added some more detail summarizing the population in the ‘characteristics of participants’ section I feel this would be sufficient e.g Ethnicity. Were participants PA levels assessed in any way?

Response: Thank you. We have removed Table 2 and added the details of ethnicity to the characteristics of participants section (see line 183 – 185). Physical activity was not assessed and a mention of this has been added to the discussion as a limitation and for future consideration, see lines 457 - 462.

7)       You reference that this qualitative work is part of a behaviour change intervention. It would be good to refer more explicitly to this intervention in the paper, especially in the next steps section of the Discussion.

Response: Thank you. We have removed reference to this work being part of a behaviour change intervention, and re-worded to be clear that this work can inform future behaviour change interventions, see lines 451-453.

8)       It could be clearer that personas were pseudonyms

Response: Thank you, we have made this clearer in the methods section.

Reviewer 2 Report

Sir, 

first at all, thank you for the opportunity to review this very interesting paper. I've found this research not only very interesting, but potentially useful in order to the planning of future interventions aimed to cope with increasing sedentarety among office workers. The methods are appropriately described, and the results are sufficiently well described. 

However, I think that some further improvements may be recommended.

First at all, I suggest the Authors to explain in the introduction whether participants to the study had previously received or not a preliminary / periodic assessment by occupational physicians. In several European countries - i.e. Italy and France, VDU workers receive a specifically designed periodic assessment of their fitness to work. As a consequence, the occupational physicians cope with the health issues of the workers, including sedentarity. As this is not extensively addressed in the interviews, some remarks about the occupational health surveillance of the workers may be useful.

Similarly, please be aware that some of the sentences you quoted suggested a very low awareness of the official recommendations/guidelines for healthy offices. Discussing such issues across the discussion section may improve the overall quality of the paper.

I have no further remarks.

Thank you again.

Author Response

Dear Reviewer 2,

We would like to take this opportunity to thank you for your positive comments regarding our paper and time that you have taken to review our work.  Please find replies to your comments below.

Best wishes

Dr Angel Chater on behalf of all co-authors

First at all, I suggest the Authors to explain in the introduction whether participants to the study had previously received or not a preliminary / periodic assessment by occupational physicians. In several European countries - i.e. Italy and France, VDU workers receive a specifically designed periodic assessment of their fitness to work. As a consequence, the occupational physicians cope with the health issues of the workers, including sedentarity. As this is not extensively addressed in the interviews, some remarks about the occupational health surveillance of the workers may be useful.

Response: Thank you. This has been added to the limitation of the study and future consideration in Line 461-462.

Similarly, please be aware that some of the sentences you quoted suggested a very low awareness of the official recommendations/guidelines for healthy offices. Discussing such issues across the discussion section may improve the overall quality of the paper.

Response: Agreed. This has been incorporated in the discussion section. Please see Line 444 - 446.